# OpenReview forum: "Reasoning as an Adaptive Defense for Safety"
_NeurIPS.cc/2025/Conference — NeurIPS 2025 poster_

### Official Review · Reviewer_a2xm · 2025-06-26

**Clarity:** 3
**Significance:** 2
**Originality:** 2
**Rating:** 4
**Confidence:** 2

**Summary:**

The paper introduces TARS, a RL framework designed to enhance the safety of LLMs by leveraging adaptive reasoning. The key idea is to train models to dynamically allocate computational resources based on the perceived complexity and potential harm of input prompts. TARS combines SFT with RL, using a mix of harmful, harmless, and ambiguous prompts to encourage nuanced reasoning. The framework employs a reward function that balances safety with task completion. Experiments demonstrate that TARS-trained models achieve better safety-refusal trade-offs, exhibit adaptive reasoning behaviors, and maintain robustness against adversarial attacks while preserving general reasoning capabilities.

**Questions:**

Please refer to "Weaknesses"

**Ethical Concerns:**

["NO or VERY MINOR ethics concerns only"]

**Final Justification:**

According to the rebuttal, my concerns have been addressed, and I will maintain my positive score for this paper.

**Limitations:**

yes

**Quality:**

3

**Strengths And Weaknesses:**

Strengths
1. This paper introduces TARS for safety in LLMs, addressing gaps in existing methods that rely on static reasoning traces or lack adaptive compute allocation.
2. This paper demonstrates improved resilience against both white-box (e.g., GCG) and black-box (e.g., PAIR) adversarial attacks compared to non-reasoning and SFT-trained baselines.
3. Comprehensive experiments validate the framework, including ablation studies, internal representation analysis, and comparisons to SFT, DPO, and RL baselines.

Weaknesses
1. Experiments are conducted on a Quen-2.5-1.5B-Instruct model; scalability to larger, state-of-the-art LLMs is not verified.
2. The multi-stage training pipeline (SFT + RL) and reliance on long chain-of-thought reasoning may increase computational overhead compared to simpler methods.
3. Evaluate TARS on additional domains (e.g., commonsense reasoning) could strengthen claims about preserving general reasoning capabilities.

---

> ### Author Rebuttal · Authors · 2025-07-31
>
> >**Opening Statement**
>
> Thank you for the detailed feedback! We conduct three additional experiments to address your comments: (1) larger scale comparison of SFT and TARS on 7B models, (2) comparison of larger existing open-weight models and SOTA circuit breaker models, and (3) evaluation of general reasoning capabilities on 5 new tasks including non-math domains. Each experiment is described below along with our responses to your comments. If you feel that our experiments address your concerns, we kindly ask that you consider improving the score. If not, please feel free to engage and ask further questions!
>
> >**Q1: Experiments are conducted on a Quen-2.5-1.5B-Instruct model; scalability to larger, state-of-the-art LLMs is not verified.**
>
> For scalability, we additionally trained 7B models via SFT/TARS (with $\lambda=0.5$) and benchmarked the safety-refusal trade-off for them. We use the same dataset and procedure as with results in the submission, but start from Qwen 2.5 7B Instruct. We also benchmarked larger open-weight models such as Llama 3.1 8B Instruct and circuit breaker models [1] which are also 7-8B (Llama-RR, Mistral-RR) to serve as strong baselines. Circuit breakers are a SOTA defense which are well known to be safe while maintaining low refusal on harmless prompts. The results are shown below as coordinates on the safety-refusal trade-off plot.
>
> **RESULTS**
> | Model               | DSR (x) | Compliance Rate (y) |
> |---------------------|--------------|--------------|
> | TARS 7B $\lambda=0.5$ |  0.970        |   0.856      |
> | SFT 7B $\lambda=0.5$ |  0.632        |   0.894      |
> | TARS 1.5B $\lambda=0.1$ | 0.814          |  0.896         |
> | TARS 1.5B  $\lambda=0.3$ |0.924          |    0.843         |
> | TARS 1.5B  $\lambda=0.5$ | 0.981       |  0.777         |
> | TARS 1.5B  $\lambda=0.7$ | 0.981        |   0.759         |
> | TARS 1.5B  $\lambda=0.9$ | 0.993          |    0.652        |
> | Llama 3.2 1B Instruct | 0.694           |   0.825         |
> | Llama 3.1 8B Instruct | 0.546            |  0.894          |
> | Llama-RR (8B) | 0.917      |     0.806       |
> | Mistral-RR (7B) | 0.922      |    0.203        |
>
> TARS on a 7B model is the furthest out on the safety-refusal frontier, outperforming SFT on the 7B scale, not to mention Llama 8B and circuit breakers, considered a SOTA approach in the community. Interestingly, TARS on a 1.5B model already beats Llama 8B and larger circuit breaker models on the safety-refusal tradeoff. And TARS on a 7B model performs better than TARS with a 1.5B model. Thus,TARS is scalable to larger models. Given our current compute budget, we will have the full comparison across all $\lambda$ ready by the camera-ready version of the paper.
>
> >**Q2: The multi-stage training pipeline (SFT + RL) and reliance on long chain-of-thought reasoning may increase computational overhead compared to simpler methods.**
>
> It is true that reasoning could use more test-time compute compared to other defenses. However, the strongest known defense methodologies mostly require training the model itself [1-3] or training a safeguard model [4]. Both become more effective as they scale in size, meaning that more training compute is required in these defenses.
>
> Fortunately, as seen in the previous response, a smaller 1.5B model that reasons can outperform larger scale 7B models. Prior work has also observed the benefits of test-time compute (i.e., reasoning) over scaling model parameters [5], showing that leveraging test-time compute on smaller models may be more efficient than the required training compute for larger models. Thus, while reasoning may use more compute than non-reasoning, the benefits may outweigh the compute requirements of reaching stronger defenses on larger scales.
>
> >**Q3: Evaluate TARS on additional domains (e.g., commonsense reasoning) could strengthen claims about preserving general reasoning capabilities.**
>
> Thank you for the feedback. We ran additional evaluations for generalization on the following benchmarks: GSM8K [6], MATH-500 [7], MMLU-Pro [8], GPQA [9], and SQuAD [10]. These cover not only math but also science and general reasoning capabilities. For MATH-500 and GSM8K, we evaluated using the ground truths by appending a system prompt that encourages the model to output answers between “\boxed{}” and performing regex matching.
>
> **RESULTS**
> | Model | GSM8K | MATH-500 | GPQA | MMLU-Pro | SQuAD | Avg. |
> |-------------|--------------------|--------------------|--------------------|--------------------|--------------------|---------|
> | TARS (Ours) | 0.5500 $\pm$ 0.0255 | 0.4246 $\pm$ 0.0122 | 0.3404 $\pm$ 0.0101 | 0.3315 $\pm$ 0.0110 | 0.9070 $\pm$ 0.0096 | **0.5107** |
>  | RL | 0.4920 $\pm$ 0.0265 | 0.4364 $\pm$ 0.0111 | 0.3020 $\pm$ 0.0088 | 0.3890 $\pm$ 0.0076 | 0.8830 $\pm$ 0.0096 | 0.5005 |
> | Base Model | 0.5330 $\pm$ 0.0341 | 0.4210 $\pm$ 0.0110 | 0.2419 $\pm$ 0.0093 | 0.3510 $\pm$ 0.0084 | 0.9230 $\pm$ 0.0096 | 0.4940 |
> | SFT | 0.0470 $\pm$ 0.0131 | 0.0162 $\pm$ 0.0032 | 0.3167 $\pm$ 0.0100 | 0.1900 $\pm$ 0.0164 | 0.4640 $\pm$ 0.0232 | 0.2068 |
> | DPO | 0.0210 $\pm$ 0.0092 | 0.0096 $\pm$ 0.0019 | 0.3040 $\pm$ 0.0099 | 0.1950 $\pm$ 0.0101 | 0.4580 $\pm$ 0.0263 | 0.1975 |
>
> TARS outperforms the base model on average with minimal degradation. We also observe that supervised methods have poor OOD generalization compared to RL, which agrees with the AIME results shown in our paper. Based on these results, TARS can preserve performance compared to the base model and sometimes even improve, showing better robustness than SFT, DPO, and no reasoning.
>
> >**REFERENCES (joint with reviewer LQBK)**
>
> [1] Zou, Andy, et al. "Improving alignment and robustness with circuit breakers." Advances in Neural Information Processing Systems 37 (2024): 83345-83373.
>
> [2] Ouyang, Long, et al. "Training language models to follow instructions with human feedback." Advances in neural information processing systems 35 (2022): 27730-27744.
>
> [3] Rafailov, Rafael, et al. "Direct preference optimization: Your language model is secretly a reward model." Advances in neural information processing systems 36 (2023): 53728-53741.
>
> [4] Inan, Hakan, et al. "Llama guard: Llm-based input-output safeguard for human-ai conversations." arXiv preprint arXiv:2312.06674 (2023).
>
> [5] Snell, Charlie, et al. "Scaling llm test-time compute optimally can be more effective than scaling model parameters, 2024." URL https://arxiv. org/abs/2408.03314 20 (2024).
>
> [6] Cobbe, Karl, et al. "Training verifiers to solve math word problems." arXiv preprint arXiv:2110.14168 (2021).
>
> [7] Hendrycks, Dan, et al. "Measuring mathematical problem solving with the math dataset." arXiv preprint arXiv:2103.03874 (2021).
>
> [8] Rein, David, et al. "Gpqa: A graduate-level google-proof q&a benchmark." First Conference on Language Modeling. 2024.
>
> [9] Wang, Yubo, et al. "Mmlu-pro: A more robust and challenging multi-task language understanding benchmark." Advances in Neural Information Processing Systems 37 (2024): 95266-95290.
>
> [10] Rajpurkar, Pranav, et al. "Squad: 100,000+ questions for machine comprehension of text." arXiv preprint arXiv:1606.05250 (2016).
>
> [11] Zhang, Yichi, et al. "Realsafe-r1: Safety-aligned deepseek-r1 without compromising reasoning capability." arXiv preprint arXiv:2504.10081 (2025).
>
> [12] Si, Shengyun, et al. "Think Before Refusal: Triggering Safety Reflection in LLMs to Mitigate False Refusal Behavior." arXiv preprint arXiv:2503.17882 (2025).
>
> [13] Zhang, Yuyou, et al. "Safety is Not Only About Refusal: Reasoning-Enhanced Fine-tuning for Interpretable LLM Safety." arXiv preprint arXiv:2503.05021 (2025).
>
> [14] Zhou, Kaiwen, et al. "The hidden risks of large reasoning models: A safety assessment of r1." arXiv preprint arXiv:2502.12659 (2025).
>
> [15] Jiang, Fengqing, et al. "Safechain: Safety of language models with long chain-of-thought reasoning capabilities." arXiv preprint arXiv:2502.12025 (2025).
>
> [16] Zhang, Yiming, et al. "Backtracking improves generation safety." arXiv preprint arXiv:2409.14586 (2024).
>
> [17] Wang, Haoyu, et al. "Safety Reasoning with Guidelines." arXiv preprint arXiv:2502.04040 (2025).
>
> [18] Setlur, Amrith, et al. "Rewarding progress: Scaling automated process verifiers for llm reasoning." arXiv preprint arXiv:2410.08146 (2024).
>
> [19] Cui, Ganqu, et al. "Process reinforcement through implicit rewards." arXiv preprint arXiv:2502.01456 (2025).
>
> [20] Wang, Teng, et al. "Towards Hierarchical Multi-Step Reward Models for Enhanced Reasoning in Large Language Models." arXiv preprint arXiv:2503.13551 (2025).

---

> > ### Author Response · Authors · 2025-08-04
> >
> > Please let us know if you had the chance to read through the rebuttals and whether your concerns are addressed. If not, please feel free to ask further questions since the discussion period concludes shortly. Otherwise, we would sincerely appreciate your consideration in raising the score. Thank you!

---

> ### Comment · Reviewer_a2xm · 2025-08-05
>
> Thank you for your reply. My concerns have been addressed, and I will maintain my positive score for this paper.

---

### Official Review · Reviewer_p1tZ · 2025-06-28

**Clarity:** 3
**Significance:** 2
**Originality:** 2
**Rating:** 4
**Confidence:** 4

**Summary:**

This paper studies how to enhance large language model (LLM) safety by post-training LLMs to enable explicit safety-aware reasoning behavior (chain-of-thought wrapped in \<think\> \</think\>) when responding to safety related questions. The proposed TARS (training adaptive reasoners for safety) recipe follows a standard SFT-then-RL post-training pipeline while highlighting some design choices regarding dataset construction, training, and reward design. For SFT, TARS distill traces of harmful prompts (4 traces for each prompt) of multiple datasets from DeepSeek-R1 without constraint on response harmfulness, and train the target model to follow reasoning patterns. For the RL stage, TARS use mixed harmful and harmless (+ambiguous) prompts and compute reward for safety, task completion, and format correctness. An additional design for the RL stage when using GRPO is that TARS compute the advantages for harmful and harmless prompts separately but update actor at once to avoid bias. Experiments show that TARS shows better safety-refusal trade-off compared to SFT, RPO, and RL without reasoning. TARS trained models show adaptive behavior by allocating more tokens on reasoning for complex safety related tasks. Moreover, TARS also add to robustness against attacks like GCG and PAIR.

**Questions:**

1. Can you provide more evaluation results on the preservation of general ability after post-training, e.g., code, factuality, etc.?
2. Can you provide a comparison between context distillation style reasoning-enhanced safety alignment and TARS?

**Ethical Concerns:**

["NO or VERY MINOR ethics concerns only"]

**Final Justification:**

The rebuttal in general addressed my concerns.

**Limitations:**

Maybe the authors can be more careful about releasing the model and data for safety concerns. Only share upon legal requests.

**Quality:**

2

**Strengths And Weaknesses:**

Strengths
- Systematically studies some design choices in safety alignment (post-training) of LLMs with explicit reasoning behaviors.
- Extensive experiments show that TARS has better safety-refusal trade-off and robustness to attacks compared to baseline SFT, DPO, and RL without reasoning.

Weakness
- Limited evaluation on preservation of LLM's general ability after TARS except for AIME.
- Missing baseline to compare with. Deliberative alignment [1] from OpenAI already demonstrated the effectiveness of safety alignment with reasoning. This paper cited DA but didn't discuss in depth the similarity and difference between the two. Even though DA is not directly reproducible since it is based on O3, it is necessary to incorporate some context distillation style baselines [2, 3] for comparison as this paper's goal is to systematically understand reasoning-enhanced safety alignment.
- The term "scaling test-time compute" is a bit abused in this paper, as in this paper it mainly means longer reasoning traces, but in general it means methods like best-of-n, beam search, and tree search at test time [4].

[1] Guan, Melody Y., et al. "Deliberative alignment: Reasoning enables safer language models." arXiv preprint arXiv:2412.16339 (2024).

[2] Snell, Charlie, Dan Klein, and Ruiqi Zhong. "Learning by distilling context." arXiv preprint arXiv:2209.15189 (2022).

[3] Askell, Amanda, et al. "A general language assistant as a laboratory for alignment." arXiv preprint arXiv:2112.00861 (2021).

[4] Snell, Charlie, et al. "Scaling llm test-time compute optimally can be more effective than scaling model parameters." arXiv preprint arXiv:2408.03314 (2024).

---

> ### Author Rebuttal · Authors · 2025-07-31
>
> >**Opening Statement**
>
> Thank you for the detailed feedback! We conduct two additional experiments to address your comments: (1) utility evaluation on 5 new tasks including non-math domains and (2) context distillation baselines following [6]. Each experiment is described below along with our responses to your comments. If you feel that our experiments address your concerns, we kindly ask that you consider improving the score. If not, please feel free to engage and ask further questions!
>
> >**Q1: Limited evaluation on preservation of LLM's general ability after TARS except for AIME.**
>
> >**Q4: Can you provide more evaluation results on the preservation of general ability after post-training, e.g., code, factuality, etc.?**
>
> Thank you for the feedback. We ran additional evaluations for generalization on the following benchmarks: GSM8K [1], MATH-500 [2], MMLU-Pro [3], GPQA [4], and SQuAD [5]. These cover not only math but also science and general reasoning capabilities. For MATH-500 and GSM8K, we evaluated using the ground truths by appending a system prompt that encourages the model to output answers between “\boxed{}” and performing regex matching.
>
> **RESULTS**
>
> | Model | GSM8K | MATH-500 | GPQA | MMLU-Pro | SQuAD | Avg. |
>  |-------------|--------------------|--------------------|--------------------|--------------------|--------------------|---------|
> | TARS (Ours) | 0.5500 $\pm$ 0.0255 | 0.4246 $\pm$ 0.0122 | 0.3404 $\pm$ 0.0101 | 0.3315 $\pm$ 0.0110 | 0.9070 $\pm$ 0.0096 | **0.5107** |
>  | RL | 0.4920 $\pm$ 0.0265 | 0.4364 $\pm$ 0.0111 | 0.3020 $\pm$ 0.0088 | 0.3890 $\pm$ 0.0076 | 0.8830 $\pm$ 0.0096 | 0.5005 |
> | Base Model | 0.5330 $\pm$ 0.0341 | 0.4210 $\pm$ 0.0110 | 0.2419 $\pm$ 0.0093 | 0.3510 $\pm$ 0.0084 | 0.9230 $\pm$ 0.0096 | 0.4940 |
> | SFT | 0.0470 $\pm$ 0.0131 | 0.0162 $\pm$ 0.0032 | 0.3167 $\pm$ 0.0100 | 0.1900 $\pm$ 0.0164 | 0.4640 $\pm$ 0.0232 | 0.2068 |
> | DPO | 0.0210 $\pm$ 0.0092 | 0.0096 $\pm$ 0.0019 | 0.3040 $\pm$ 0.0099 | 0.1950 $\pm$ 0.0101 | 0.4580 $\pm$ 0.0263 | 0.1975 |
>
> TARS outperforms the base model on average with minimal degradation. We also observe that supervised methods have poor OOD generalization compared to RL, which agrees with the AIME results shown in our paper. These results show that TARS can preserve performance compared to the base model and sometimes even improves, showing better robustness than SFT, DPO, and no reasoning.
>
> >**Q2: Missing baseline to compare with. Deliberative alignment [1] from OpenAI already demonstrated the effectiveness of safety alignment with reasoning. This paper cited DA but didn't discuss in depth the similarity and difference between the two. Even though DA is not directly reproducible since it is based on O3, it is necessary to incorporate some context distillation style baselines [2, 3] for comparison as this paper's goal is to systematically understand reasoning-enhanced safety alignment.**
>
> >**Q5: Can you provide a comparison between context distillation style reasoning-enhanced safety alignment and TARS?**
>
> Certainly! Based on your feedback, we ran additional SFT baselines that use context distillation following [6]. We adopt their exact guidelines to distill self-reflecting and self-refining reasoning traces from gpt-4.1-mini and also do rejection sampling. We then train through SFT without the guidelines to distill context. We also compare RealSafe-R1-1.5B [7] and SafeChain-8B [8] which were trained for safety using context distillation and rejection sampling. The results are shown below as coordinates on the safety-refusal trade-off plot.
>
> **RESULTS**
> | Model  | DSR (x) | Compliance Rate (y) |
>  |---------------------|---------|---------------------|
> | SFT with guidelines $\lambda=0.1$ | 0.828  | 0.614  |
> | SFT with guidelines $\lambda=0.3$ | 0.890  | 0.597  |
> | SFT with guidelines $\lambda=0.5$ | 0.887  | 0.576  |
> | SFT with guidelines $\lambda=0.7$ | 0.890  | 0.573  |
> | SFT with guidelines $\lambda=0.9$ | 0.881  | 0.510  |
> | RealSafe-R1-1.5B | 0.973  | 0.090  |
> | SafeChain-8B | 0.686  | 0.937  |
> | TARS $\lambda=0.1$ | 0.814          |  0.896         |
> | TARS $\lambda=0.3$ | 0.924          |    0.843         |
> | TARS $\lambda=0.5$ | 0.981       |  0.777         |
> | TARS $\lambda=0.7$ | 0.981        |   0.759         |
> | TARS $\lambda=0.9$ | 0.993          |    0.652        |
>
> We see that TARS achieves the best safety-refusal trade-off compared to SFT methods that use context distillation. In detail, curating SFT traces with guidelines moderately improves upon safety compared to normal SFT but still remains less safe than TARS. RealSafe-R1 and SafeChain also cannot become safer while maintaining low refusal. Although there are many ways to curate reasoning traces for SFT, we believe that the major limitation of SFT is its poor performance out-of-distribution (as also shown for reasoning in math/code, where SFT is often known to memorize while RL generalizes), leading to these results. On the contrary, RL based methods excel out of distribution. We will include these results in Section 5 of the paper.
>
> >**Q3: The term "scaling test-time compute" is a bit abused in this paper, as in this paper it mainly means longer reasoning traces, but in general it means methods like best-of-n, beam search, and tree search at test time [4].**
>
> We will clarify throughout the paper that we refer to a specific way of scaling test-time compute by generating longer reasoning traces or chain-of-thought traces before producing an answer, akin to DeepSeek-R1 (“R1-style”) [9].
>
> >**REFERENCES**
>
> [1] Cobbe, Karl, et al. "Training verifiers to solve math word problems." arXiv preprint arXiv:2110.14168 (2021).
>
> [2] Hendrycks, Dan, et al. "Measuring mathematical problem solving with the math dataset." arXiv preprint arXiv:2103.03874 (2021).
>
> [3] Rein, David, et al. "Gpqa: A graduate-level google-proof q&a benchmark." First Conference on Language Modeling. 2024.
>
> [4] Wang, Yubo, et al. "Mmlu-pro: A more robust and challenging multi-task language understanding benchmark." Advances in Neural Information Processing Systems 37 (2024): 95266-95290.
>
> [5] Rajpurkar, Pranav, et al. "Squad: 100,000+ questions for machine comprehension of text." arXiv preprint arXiv:1606.05250 (2016).
>
> [6] Wang, Haoyu, et al. "Safety Reasoning with Guidelines." arXiv preprint arXiv:2502.04040 (2025).
>
> [7] Zhang, Yichi, et al. "Realsafe-r1: Safety-aligned deepseek-r1 without compromising reasoning capability." arXiv preprint arXiv:2504.10081 (2025).
>
> [8] Jiang, Fengqing, et al. "Safechain: Safety of language models with long chain-of-thought reasoning capabilities." arXiv preprint arXiv:2502.12025 (2025).
>
> [9] Guo, Daya, et al. "Deepseek-r1: Incentivizing reasoning capability in llms via reinforcement learning." arXiv preprint arXiv:2501.12948 (2025).

---

> ### Comment · Reviewer_p1tZ · 2025-08-03
>
> Thank you for your rebuttal. My concern on general ability preservation is addressed. However, I still have concerns about the relationship and difference between TARS and deliberative alignment (DA). As you mentioned "curating SFT traces with guidelines moderately improves upon safety compared to normal SFT", it means that context distillation is indeed helpful (which is the first stage in DA), and since this is not an apple-to-apple comparison (you only compared SFT with TARS, no follow-up RL stage), it is possible that DA is better than TARS with a complete post-training pipeline. The essence here is that context distillation from explicitly defined policy potentially helps the grounding in the reasoning trace that helps models to better reason about safety, which is orthogonal to post-training strategies.

---

> > ### Author Response · Authors · 2025-08-05
> >
> > Thank you for the response. To address your concern, we ran an experiment combining context distillation with RL in the style of Deliberative Alignment (DA) and compared it with TARS. We find that TARS outperforms context distillation on the safety-refusal trade-off due to less refusal. We also elaborate on the relationship between TARS and context distillation to illustrate their orthogonality in tackling safety training.
> >
> > **Experimental setup**: For a more fair comparison between context distillation and TARS, we follow DA and use guidelines as context in the reward model during online RL. We do this by first changing the reward model to gpt-4.1-mini then providing separate rubrics for harmful and harmless prompts which measure scores among $\\{0, 0.5, 1\\}$. This helps the model learn to refuse on harmful prompts and also learn specific answers by trying to fully answer harmless prompts. We train a model for $\lambda=0.5$ on the same prompts as TARS.
> >
> > **Results and findings:** The results are shown below as coordinates on the safety-refusal trade-off plot. For an intuitive metric on safety-refusal optimality, we also provide the geometric mean ($\sqrt{x\times y}$) of each model, which represents the outwardness on the Pareto frontier of safety-refusal plots (see Figure 2 as an example). A higher outwardness score means a better trade-off between safety and compliance.
> >
> > | Model | DSR (x) | Compliance Rate (y) | Geometric Mean ($\sqrt{x\times y}$)
> > |---------------------|--------------|--------------|---|
> > | Context Distillation $\lambda=0.5$ |  0.989     |   0.668   |  0.813  |
> > | TARS $\lambda=0.1$ | 0.814      |  0.896       |0.854|
> > | TARS $\lambda=0.3$ | 0.924      |    0.843     |0.883|
> > | TARS $\lambda=0.5$ | 0.981     |  0.777       |0.872|
> > | TARS $\lambda=0.7$ | 0.981      |   0.759       |0.863|
> > | TARS $\lambda=0.9$ | 0.993      |    0.652      |0.805|
> >
> > TARS is generally more optimal on the outwardness score than DA-style context distillation on the safety-refusal trade-off with TARS $\lambda=0.5$ having a geometric mean of 0.872 while using context distillation on $\lambda=0.5$ achieves 0.813. Context distillation refuses much more, on par with TARS $\lambda=0.9$, which is clearly a suboptimal choice of $\lambda$ for TARS.
> >
> > **General comments on the relationship between TARS and context distillation:** Context distillation in the style of DA is generally used as a data curation strategy or LLM judge prompting strategy. It can be used together with TARS in various ways (e.g., as a system prompt during rollout) because the ingredients of TARS does not restrict on specific curation methods or prohibit using guidelines in the reward model. TARS only prescribes that the base model is trained lightly with SFT and the reward system is separated. These prescriptions made by TARS provide guidelines on the training strategy as opposed to DA, which prescribes the input/outputs to the models involved. Thus, TARS and context distillation may be viewed as tackling orthogonal pieces of the safety training problem.
> >
> > That said, we also hypothesize two reasons/limitations why context distillation in the reward model does worse compared to TARS.
> >
> > 1. While providing context may help ground reasoning traces to be more related to safety, guidelines are typically used for discrete rubrics. This limits the reward signal’s ability to incentivize adaptive reasoning for better answers. On the other hand, TARS allows continuous rewards in both safety and task-completion providing better signals for reasoning, which is the one we adopt. In our training run, we actually found that using context distillation leads to shorter reasoning traces over the course of training compared to TARS, which could explain the poor performance on the safety-refusal trade-off.
> >
> > 2. LLMs have over-refusal mechanisms that make it hard to leverage context distillation specifically for safety. Since context distillation requires using an instruction-tuned LLM, any request that contains harmful content will be refused. This applies not just to malicious requests but also the judgement of malicious (request, answer) pairs. Specifically, we had to create a workaround for this case by manually setting the score to 0. To bypass this problem, one would have to train their own reward model that complies to harmful requests, likely a large model for sufficient capabilities, but at that point, doing TARS is more efficient.
> >
> > *Please let us know if this response and our results address your concerns. We will add this discussion to the paper and are also happy to answer any remaining questions.*

---

> ### Comment · Reviewer_p1tZ · 2025-08-05
>
> Thank you for the additional results and discussions. Please add the discussion to the final paper (maybe a more comprehensive version) to make the paper more solid and meaningful for the community and future research. I will raise my score.

---

### Official Review · Reviewer_LQBK · 2025-06-30

**Clarity:** 3
**Significance:** 2
**Originality:** 2
**Rating:** 4
**Confidence:** 3

**Summary:**

The authors propose a reinforcement learning methods to train models against safety vulnerabilities, called Training Adapative Reasoners for Safety（TARS） with three key designs: 1. imperfect SFT training 2. combing safety reasoning data whose reasoning traces distilled from R1 and general utility data 3. safe and reasoning reward designs. With their proposed method, the authors can improve Qwen-2.5-1.5B-instruct's safety against traditional DPO and SFT methods.

**Questions:**

1. You say adding utility data are try to avoid overfitting on "Sorry, I cannot" in RL stage. But if we adopt dense reward to judge whether LLMs reasoning process is correct or proper, can we avoid overfitting without utility data?

2. Why use adopt UltraFeedback as utility data? Why not use GSM8K or LIMO? They are reasoning types and may make LLMs converge better?

**Ethical Concerns:**

["NO or VERY MINOR ethics concerns only"]

**Final Justification:**

The authors' rebuttal addressed my concerns. I decide to raise my score.

**Limitations:**

Yes

**Quality:**

2

**Strengths And Weaknesses:**

Strengths:

1. The papers writing is clear.

2. The method is simple yet effective.

Weaknesses:

1. The paper is lack of novelty. Extracting reasoning traces for safe data is not new. Although former works (as authors cite in their paper) only focus on SFT, I cannot agree adding an RL procedure after SFT is an important contribution.

2. Whether enabling safe reasoning abilities in general models (instead of reasoning models) are good need more discussion. As under different attacks, Claude and Llama are usually performs better than reasoning models. And reasoning models has their own attacks like H-CoT. Since the authors does not aim to improve LRMs safety like many other papers (like SafeChain) does, I think they need more discussion on why we need reasoning to enhance safety.

3.The evaluations are strange. The baseline methods are just SFT and DPO, I think the results are not surprising, SFT+RL>SFT or RLHF which is demonstrated by R1. I suggest authors add their other SFT methods with their proposed datasets (like SFT with SafeChain or RealSafeR1).

4.Evaluations on utility is also wired. They say they are not trained for mathematical problems, but they only use AIME as baseline. And I do not agree that the model cannot output mathematical types as Llama and Qwen's instruction model can achieve high score on GSM8K and MathQA, the authors only need to modify their prompt. Therefore, I suggest the authors use the ground truth to evaluate models reasoning ability as they are simple and correct, LLM-as-a-judge may exist bias and are hardly used as reasoning evaluators.

---

> ### Author Rebuttal · Authors · 2025-07-31
>
> >**Opening Statement**
>
> Thank you for your thorough feedback! We believe there might be a misunderstanding in our contributions and would like to clarify them. While our results may be intuitive, to the best of our knowledge, ours is the first open-source work showing how to apply reasoning through online RL for safety. We believe that a simple recipe that improves safety is of interest to the NeurIPS community, irrespective of proprietary, closed-source models. This is analogous to how 8B Qwen/Llama models offer significant value despite being behind GPT/Gemini models. Also, we conduct four additional experiments to address your comments: (1) comparison of SafeChain, (2) comparison of larger open-weight models and SOTA circuit breaker models, (3) several SFT experiments with better curation strategies including RealSafe-R1, and (4) utility evaluation on 5 new tasks including ground truth evaluation on GSM8K and MATH-500. If you feel that our experiments address your concerns, we kindly ask that you consider improving the score. If not, please feel free to engage and ask further questions!
>
>
> >**Q1: The paper is lack of novelty. Extracting reasoning traces for safe data is not new. Although former works (as authors cite in their paper) only focus on SFT, I cannot agree adding an RL procedure after SFT is an important contribution.**
>
> Our contributions are not the distillation of reasoning traces or online RL after SFT, but rather
> 1. a concrete recipe with 3 key ingredients (Section 3) that improves the safety-refusal trade-off backed up with substantial ablations (Section 7)
> 2. the identification of a type of reward hacking in the form of reasoning degeneration that is unique to open-domain non-rule based RL (Section 3: Desiderata, Figure 5(c))
> 3. a fair comparison to SFT/DPO and RL without reasoning using the same compute/data budgets.
>
> Our work is not just another RL reasoning paper, but a distinct adaptation of online RL to the open-domain of safety, which uncovers and addresses the problem of reasoning degeneration. Furthermore, while most papers compare already trained models, we conduct controlled experiments by removing confounding factors (compute/data) to directly compare different training methods (lines 225-226). We believe that our work helps resolve conflicting results in the community by providing a fair comparison on CoT safety training [11-14]. Nonetheless, we agree that these points can be made clearer and will make sure to clarify them in the revised paper.
>
> >**Q2: Whether enabling safe reasoning abilities in general models (instead of reasoning models) are good need more discussion. As under different attacks, Claude and Llama are usually performs better than reasoning models. And reasoning models has their own attacks like H-CoT. Since the authors does not aim to improve LRMs safety like many other papers (like SafeChain) does, I think they need more discussion on why we need reasoning to enhance safety.**
>
> Thank you for the question! We might be misinterpreting your question, so please correct us if our response is insufficient.
>
> First, for a more rigorous argument on why we need reasoning to enhance safety, we conduct additional experiments on comparing the following.
> 1. SafeChain [15], using their released 8B model.
> 2. Larger open-weight models such as Llama 3.1 8B Instruct.
> 3. Circuit breakers (Llama-RR, Mistral-RR) [1], a SOTA defense which adaptively refuses to harmful prompts and is helpful on harmless prompts.
>
> The results are shown below as coordinates on the safety-refusal trade-off plot.
>
> **RESULTS**
>
> |Model|DSR (x)|Compliance Rate (y)|
> |---|---|---|
> |SafeChain-8B|0.686|0.937|
> |Llama 3.2 1B Instruct|0.694|0.825|
> |Llama 3.1 8B Instruct|0.546|0.894|
> |Llama-RR (8B)|0.917|0.806|
> |Mistral-RR (7B)|0.922|0.203|
> |TARS $\lambda=0.1$|0.814|0.896|
> |TARS $\lambda=0.3$|0.924|0.843|
> |TARS $\lambda=0.5$|0.981|0.777|
> |TARS $\lambda=0.7$|0.981|0.759|
> |TARS $\lambda=0.9$|0.993|0.652|
>
> We see that TARS maintains the best safety-refusal trade-off compared to all three model types. Specifically, SafeChain which enhances the safety of LRMs are more compliant but have lower safety. TARS can even outperform a larger Llama 8B model and Llama-RR, which is an additional layer of safety.
>
> These results show that training models to reason for safety through TARS may be a better method than training off-the-shelf reasoning models or SOTA defenses.
>
> Second, we note that *reasoning is particularly valuable for safety* because it enables the model to dynamically allocate test-time compute to better assess each prompt individually. This is especially important for ambiguous or edge-case prompts, where determining the safest and most appropriate response may depend on subtle contextual cues. By adaptively reasoning rather than relying on a fixed, one-size-fits-all model, models can navigate the safety–refusal trade-off better, leading to improved generalization across diverse safety scenarios.
>
> We will add these valuable discussions to our paper.
>
> >**Q3: The evaluations are strange. I suggest authors add their other SFT methods with their proposed datasets (like SFT with SafeChain or RealSafeR1).**
>
> Thank you for the feedback. Please note that despite the apples-to-oranges comparisons that exist, the methods we compare are representative in the field of reasoning for safety (e.g., DPO is based on [16]). Based on your suggestion, we additionally benchmark 3 more SFT methods.
>
> 1. Curating SFT reasoning traces to self-reflect and self-refine through guidelines [17] with their proposed prompting.
> 2. Using rejection sampling (RS) with the same reward system as TARS.
> 3. RealSafe-R1 [11] and SafeChain [15], using their released models.
>
> The results are shown below as coordinates on the safety-refusal trade-off plot.
>
> **RESULTS**
> |Model|DSR (x)|Compliance Rate (y)|
> |---|---|---|
> |SFT guidelines $\lambda=0.1$|0.828|0.614|
> |SFT guidelines $\lambda=0.3$|0.890|0.597|
> |SFT guidelines $\lambda=0.5$|0.887|0.576|
> |SFT guidelines $\lambda=0.7$|0.890|0.573|
> |SFT guidelines $\lambda=0.9$|0.881|0.510|
> |SFT RS $\lambda=0.1$|0.771|0.654|
> |SFT RS $\lambda=0.3$|0.787|0.643|
> |SFT RS $\lambda=0.5$|0.813|0.639|
> |SFT RS $\lambda=0.7$|0.782|0.608|
> |SFT RS $\lambda=0.9$|0.795|0.566|
> |RealSafe-R1-1.5B|0.973|0.090|
> |SafeChain-8B|0.686|0.937|
> |TARS $\lambda=0.1$|0.814|0.896|
> |TARS $\lambda=0.3$|0.924|0.843|
> |TARS $\lambda=0.5$|0.981|0.777|
> |TARS $\lambda=0.7$|0.981|0.759|
> |TARS $\lambda=0.9$|0.993|0.652|
>
> Overall, we see that TARS achieves the best safety-refusal trade-off compared to the three SFT methods. RealSafe-R1 is safe at the expense of compliance and SafeChain has lower safety. Although there are many ways to curate reasoning traces for SFT, we believe that the major limitation of SFT is its poor performance out-of-distribution, leading to these results. On the contrary, RL based methods excel out of distribution. We will include these results in Section 5 of the paper.
>
> >**Q4: Evaluations on utility are also weird. Therefore, I suggest the authors use the ground truth to evaluate models’ reasoning ability.**
>
> Based on your suggestion, we ran additional experiments for GSM8K [6] and MATH-500 [7] on ground-truth evaluations. We appended a system prompt to encourage output answers to be between “\boxed{}” and performed regex matching for evaluation. We also benchmarked additional datasets that are not math: GPQA [8], MMLU-Pro [9], and SQuAD [10].
>
> **RESULTS**
>
> | Model | GSM8K | MATH-500 | GPQA | MMLU-Pro | SQuAD | Avg. |
>  |---|---|---|---|---|---|---|
> | TARS (Ours) | 0.5500 $\pm$ 0.0255 | 0.4246 $\pm$ 0.0122 | 0.3404 $\pm$ 0.0101 | 0.3315 $\pm$ 0.0110 | 0.9070 $\pm$ 0.0096 | **0.5107** |
>  | RL | 0.4920 $\pm$ 0.0265 | 0.4364 $\pm$ 0.0111 | 0.3020 $\pm$ 0.0088 | 0.3890 $\pm$ 0.0076 | 0.8830 $\pm$ 0.0096 | 0.5005 |
> | Base Model | 0.5330 $\pm$ 0.0341 | 0.4210 $\pm$ 0.0110 | 0.2419 $\pm$ 0.0093 | 0.3510 $\pm$ 0.0084 | 0.9230 $\pm$ 0.0096 | 0.4940 |
> | SFT | 0.0470 $\pm$ 0.0131 | 0.0162 $\pm$ 0.0032 | 0.3167 $\pm$ 0.0100 | 0.1900 $\pm$ 0.0164 | 0.4640 $\pm$ 0.0232 | 0.2068 |
> | DPO | 0.0210 $\pm$ 0.0092 | 0.0096 $\pm$ 0.0019 | 0.3040 $\pm$ 0.0099 | 0.1950 $\pm$ 0.0101 | 0.4580 $\pm$ 0.0263 | 0.1975 |
>
> TARS outperforms the base model on average with minimal degradation. We also observe that supervised methods have poor OOD generalization compared to RL, which agrees with the AIME results. We will modify our claim on generalization (as it is a supplementary finding in our work) and state that TARS can preserve performance compared to the base model and sometimes even improves.
>
> >**Q5: You say adding utility data are try to avoid overfitting on "Sorry, I cannot" in RL stage. But if we adopt dense reward to judge whether LLMs reasoning process is correct or proper, can we avoid overfitting without utility data?**
>
> Yes. The denser the reward becomes, the less reward hacking there will be. In fact, an oracle reward would have no reward hacking. However, there is no such reward system for open-domain non-rule based tasks, which is why we utilize Moderation from OpenAI and mix in harmless prompts. While building such a reward system is a critical next step to investigate, we also believe that given the efficacy of just pure outcome rewards and instabilities of tuning associated with PRMs [18-20], having a pure outcome-reward based approach for safety is a good starting point.
>
> >**Q6: Why use adopt UltraFeedback as utility data? Why not use GSM8K or LIMO? They are reasoning types and may make LLMs converge better?**
>
> That is a great question! While we also tried solely mixing in math datasets, we found that it helps less than general reasoning capabilities. We suspect that broader reasoning on harmless prompts covers more topics of safety prompts than just math prompts, making it better for reasoning to transfer over. We will add this in the discussion!
>
> >**REFERENCES**
>
> (Due to lack of space, please see reviewer a2xm’s reference list)

---

> > ### Comment · Reviewer_LQBK · 2025-08-01
> >
> > Thanks for your rebuttal. However, I still have some concerns on your work.
> >
> > 1. What is the total training cost for you method? Can you use LoRA fine-tune? As circuit breaker adopts LoRA, its training cost is really low. Can you compare its efficiency?
> >
> > 2. Do we need to add more data if we want to align larger LLMs.
> >
> > 3. How to select the best $\lamda$ for satisfying compliance and DSR? What is the proper DSR and compliance in your mind?

---

> > > ### Author Response · Authors · 2025-08-04
> > >
> > > >**Overview**
> > >
> > > Thank you for responding! We clarify training efficiency of TARS in comparison with circuit breakers by training a TARS model with LoRA, and find that in a nutshell, TARS with LoRA is still more effective than circuit breakers on the safety-refusal frontier. We also demonstrate that TARS does not quite need more data for training a larger LLM since our 7B model trained on the very same dataset as the 1.5B model performs the best on the frontier.  Finally, we elaborate on the choice of $\lambda$, and how practitioners can potentially go about choosing it. Please let us know if this response addresses your concerns, and we are also happy to discuss further.
> > >
> > > >**What is the total training cost for you method? Can you use LoRA fine-tune? As circuit breaker adopts LoRA, its training cost is really low. Can you compare its efficiency?**
> > >
> > > This is a great question on training efficiency! For a fair comparison on efficiency, we train a 7B TARS $\lambda=0.5$ model starting from Qwen 2.5 7B Instruct following our approach. We train one model with full fine-tuning and one model with LoRA to compare its performance. We use the same rank ($r=16$) as used by the circuit breakers approach. The results are shown below as coordinates on the safety-refusal trade-off plot. For an intuitive metric on safety-refusal optimality, we also provide the geometric mean ($\sqrt{x\times y}$) of each model, which represents the outwardness on the Pareto frontier of safety-refusal plots (see Figure 2 as an example). A higher outwardness score means a better trade-off between safety and compliance.
> > >
> > > | Model               | DSR (x) | Compliance Rate (y) | Geometric Mean ($\sqrt{x\times y}$)
> > > |---------------------|--------------|--------------|---|
> > > | TARS 7B $\lambda=0.5$ |  0.970        |   0.856      |0.912|
> > > | TARS 7B LoRA $\lambda=0.5$ |  0.911        |   0.878      |0.894|
> > > | TARS 1.5B $\lambda=0.1$ | 0.814      |  0.896       |0.854|
> > > | TARS 1.5B  $\lambda=0.3$ |0.924      |    0.843     |0.883|
> > > | TARS 1.5B  $\lambda=0.5$ | 0.981     |  0.777       |0.872|
> > > | TARS 1.5B  $\lambda=0.7$ | 0.981      |   0.759       |0.863|
> > > | TARS 1.5B  $\lambda=0.9$ | 0.993      |    0.652      |0.805|
> > > | Llama-RR (8B) | 0.917      |     0.806       |0.861|
> > > | Mistral-RR (7B) | 0.922      |    0.203        |0.433|
> > >
> > > First, we note that full fine-tuned TARS 7B $\lambda=0.5$ is the most optimal with a geometric mean of 0.912. Second, while TARS 7B $\lambda=0.5$ with LoRA slightly loses performance, it is still more optimal than both circuit breaker models ($0.894 > 0.861, 0.433$). Thus, on the same model scale, TARS with LoRA can outperform circuit breakers with LoRA.
> > >
> > > >**Do we need to add more data if we want to align larger LLMs.**
> > >
> > > In the previous response, we trained a 7B model with TARS by using exactly the same procedure, the same dataset, and same number of training epochs. Yet, TARS 7B is more optimal on the safety-refusal trade-off compared to TARS 1.5B. Therefore, dataset size is not a major issue when scaling to larger models, at least up to the scale we attempted.
> > >
> > > >**How to select the best $\lambda$ for satisfying compliance and DSR? What is the proper DSR and compliance in your mind?**
> > >
> > > In general, we defaulted to using $\lambda=0.5$ in our experiments for the most part as it is a natural first choice. However, to select $\lambda$, a practitioner can use the geometric mean as it represents the safety-refusal optimality. Among the ratios we computed for TARS 1.5B, $\lambda=0.3$ is the best choice with 0.883. The good aspect about our method is that it allows leniency in controlling the trade-off, and one can also utilize other statistics to decide which $\lambda$ to use. Someone that values safety at the expense of losing some compliance on harmless prompts will choose a slightly higher $\lambda$ while someone who wants a model to be more helpful will choose a lower $\lambda$.

---

> > > > ### Comment · Reviewer_LQBK · 2025-08-04
> > > >
> > > > Thanks for your response. I have no further concerns and will increase my score.

---

### Official Review · Reviewer_EJhw · 2025-07-05

**Clarity:** 3
**Significance:** 2
**Originality:** 2
**Rating:** 4
**Confidence:** 5

**Summary:**

This paper introduces TARS, a reinforcement learning framework designed to improve how large language models reason about safety. The method follows a three-stage training process: it begins with a lightly trained supervised fine-tuning phase, followed by the construction of a prompt set that mixes harmful and ambiguous harmless examples, and concludes with online RL using distinct reward models for safety and helpfulness. A key idea is to encourage the model to engage in chain-of-thought reasoning before deciding whether to answer or refuse. Empirical results show that TARS achieves better separation between harmful and ambiguous prompts and strikes a stronger balance between refusing unsafe inputs and completing helpful ones, compared to standard SFT and baseline RL approaches.

**Questions:**

Please see Weakness part.

**Ethical Concerns:**

["NO or VERY MINOR ethics concerns only"]

**Final Justification:**

I keep my score.

**Limitations:**

Please see Weakness part.

**Quality:**

3

**Strengths And Weaknesses:**

Strengths:
1. The paper is very well written, with clear structure and logical flow throughout.

2. The authors conduct thorough ablation studies, which help isolate the contribution of each component in the proposed method.


Weaknesses:
1. The author mentioned that, as shown in Figure 2, the proposed RL method outperforms the SFT baseline. However, I believe this is not a sufficiently strong piece of evidence, as the additional SFT data used by the authors was not carefully curated. Therefore, the full potential of SFT was not fully explored. For example, the related work [44] cited by the authors uses specific guidelines to construct the SFT dataset. Moreover, the experiments were only conducted on a 1.5B model, which further limits the scope of the study. I understand that the authors may not have access to sufficient computational resources to conduct larger-scale experiments. Therefore, I suggest that the authors consider softening the claims made in the paper accordingly.

2. About reward design in RL: After reading this part, I still find it unclear why adding harmless prompts and separating the reward models effectively mitigates reward hacking in safety reasoning. In theory, the model could simply learn to distinguish between different types of prompts and adopt distinct response strategies (e.g., direct refusal for harmful prompts, long-form reasoning for helpful ones).
       It’s possible that the limited capacity of the small-scale model used in the experiments constrains such behavior, or that the inclusion of ambiguous prompts plays a key role in mitigating reward hacking.
      To clarify this point, I suggest the authors include an additional experiment comparing the response lengths for ambiguous prompts, and present the results in Figure 5(c).

3. About incomplete sft: how to determine or define incomplete sft, how to determine the hyperparameter choices of  incomplete sft.

---

> ### Author Rebuttal · Authors · 2025-07-31
>
> >**Opening Statement**
>
> Thank you for the detailed feedback! We conduct three additional experiments to address your comments: (1) three SFT experiments with better curation strategies including guidelines [44], (2) evaluation of generation length on ambiguous prompt over training, and (3) evaluation of model diversity to quantify “incomplete SFT”. Each experiment is described below along with our responses to your comments. If you feel that our experiments address your concerns, we kindly ask that you consider improving the score. If not, please feel free to engage and ask further questions!
>
>
> >**Q1: The author mentioned that, as shown in Figure 2, the proposed RL method outperforms the SFT baseline. However, I believe this is not a sufficiently strong piece of evidence, as the additional SFT data used by the authors was not carefully curated. Therefore, the full potential of SFT was not fully explored. For example, the related work [44] cited by the authors uses specific guidelines to construct the SFT dataset. Moreover, the experiments were only conducted on a 1.5B model, which further limits the scope of the study. I understand that the authors may not have access to sufficient computational resources to conduct larger-scale experiments. Therefore, I suggest that the authors consider softening the claims made in the paper accordingly.**
>
> Thank you for pointing this out. To address your concern, we conduct comparisons on three different SFT based methods including the usage of guidelines.
> 1. Using guidelines to self-reflect and self-refine in the reasoning by following the proposed method [44].
> 2. Using rejection sampling (RS) with the same reward system as TARS.
> 3. RealSafe-R1 [1] and SafeChain [2], which also use guidelines and rejection sampling to curate data.
>
> Rejection sampling offers a stronger baseline because it curates reasoning traces that are more likely to refuse only to harmful prompts. Furthermore, since we use the same reward system as TARS, it becomes a direct comparison between a strong approach for SFT and TARS. For RealSafe-R1 and SafeChain, we directly evaluate their released models: RealSafe-R1-1.5B and SafeChain-8B. For replicating [44], we use their proposed guideline prompts and curate reasoning traces through gpt-4.1-mini. The results are shown below as coordinates on the safety-refusal trade-off plot.
>
> **RESULTS**
>
> | Model  | DSR (x) | Compliance Rate (y) |
>  |-----|----|----|
> | SFT with guidelines $\lambda=0.1$ | 0.828  | 0.614  |
> | SFT with guidelines $\lambda=0.3$ | 0.890  | 0.597  |
> | SFT with guidelines $\lambda=0.5$ | 0.887  | 0.576  |
> | SFT with guidelines $\lambda=0.7$ | 0.890  | 0.573  |
> | SFT with guidelines $\lambda=0.9$ | 0.881  | 0.510  |
> | SFT RS $\lambda=0.1$ | 0.771  | 0.654  |
> | SFT RS $\lambda=0.3$ | 0.787  | 0.643 |
> | SFT RS $\lambda=0.5$ | 0.813  | 0.639  |
> | SFT RS $\lambda=0.7$ | 0.782  |0.608  |
> | SFT RS $\lambda=0.9$ | 0.795  | 0.566  |
> | RealSafe-R1-1.5B | 0.973  | 0.090  |
> | SafeChain-8B | 0.686  | 0.937  |
> | TARS $\lambda=0.1$ | 0.814          |  0.896         |
> | TARS $\lambda=0.3$ | 0.924          |    0.843         |
> | TARS $\lambda=0.5$ | 0.981       |  0.777         |
> | TARS $\lambda=0.7$ | 0.981        |   0.759         |
> | TARS $\lambda=0.9$ | 0.993          |    0.652        |
>
> Overall, we see that TARS achieves the best safety-refusal trade-off compared to carefully curated reasoning traces for SFT. In detail, curating SFT traces with guidelines moderately improves upon safety compared to normal SFT but still remains less safe than TARS. Curating traces with rejection sampling remains in the vicinity of normal SFT on the safety-refusal trade-off. RealSafe-R1-1.5B is “really safe” but at the expense of compliance and SafeChain-8B has lower safety. Although there are many ways to curate reasoning traces for SFT, we believe that the major limitation of SFT is its poor performance out-of-distribution, leading to these results, whereas RL based methods generalize better, akin to what we see in other domains in reasoning [3,4]. We will include these results in the paper and soften our claim.
>
> >**Q2: About reward design in RL: After reading this part, I still find it unclear why adding harmless prompts and separating the reward models effectively mitigates reward hacking in safety reasoning. In theory, the model could simply learn to distinguish between different types of prompts and adopt distinct response strategies (e.g., direct refusal for harmful prompts, long-form reasoning for helpful ones).
> It’s possible that the limited capacity of the small-scale model used in the experiments constrains such behavior, or that the inclusion of ambiguous prompts plays a key role in mitigating reward hacking.
> To clarify this point, I suggest the authors include an additional experiment comparing the response lengths for ambiguous prompts, and present the results in Figure 5(c).**
>
> This is a great question! While in principle the model could definitely learn to separate behavior on various prompts, we do not see this behavior emerging. We suspect that this is because optimization during RL is not perfect and often in the “underfitting” regime (as usual), such that the model learns to often correlate patterns in model rollouts with reward. For example, when mixing in harmless prompts that naturally admit longer reasoning traces, the model can learn to preserve longer length rollouts overall since longer traces are expected to correlate with higher reward. Of course, we do not expect this trend to continue when training RL until convergence in *training reward*, but our training (and most RL training for LLMs) generally operates far away from this regime. Therefore, by mixing in a broad coverage of harmless prompts through UltraFeedback, we are forcing the model to learn specific related answers to each prompt. This constrains the model's tendency to learn generic refusals and encourages prompt-specific behavior.
>
> Based on your suggestion for Figure 5(c), we evaluate the response length on ambiguous prompts during training without harmless prompts. The results are that the ambiguous prompts show a similar response length decline to the harmful prompts over training, hitting 0 response length around 50 training steps. Although ambiguous prompts help improve the safety-refusal trade-off, they are not able to fully address reasoning degeneration and clearly the RL training needs a different anchor to pick upon for preserving reasoning. We will add this point to the paper.
>
> >**Q3: About incomplete sft: how to determine or define incomplete sft, how to determine the hyperparameter choices of incomplete sft.**
>
> We apologize for the confusion. By "incomplete SFT", we are not referring to a specific hyperparameter setting but rather a lightweight trained SFT model that can still sample fairly diverse traces during RL. To pick the right checkpoint for SFT,  we measured the response diversity of $\pi_\mathrm{SFT}$ trained under different hyperparameters. Specifically, we measured the average and maximum score out of 8 different generations on the harmless training prompts used for RL (i.e., "Best-of-8" and "Avg-of-8" scores under our reward model). This is analogous to pass@k scores for binary evaluations that are typically used for quantifying diversity of model rollouts during RL. We tried the following settings.
> 1. Setting 1 - Learning Rate: 1e-5, Epochs: 3
> 2. Setting 2 - Learning Rate: 3e-5, Epochs: 3
> 3. Setting 3 - Learning Rate: 1e-4, Epochs: 6
>
> **Diversity Results**
> | Setting | 1 | 2 | 3 |
> |---------|--------|--------|--------|
> | Avg-of-8 | 0.893 | **0.912** | 0.849 |
> | Best-of-8 |0.947 | **0.950** | 0.912 |
>
> We see that setting 2 has the highest diversity, which is the learning rate and epochs we adopt in TARS. A lightly trained SFT model with diverse enough rollouts that also samples high-rewarding traces is our goal. We will change the phrasing from “incomplete SFT” to “lightweight SFT” for clarity, and add this discussion to the paper.
>
>
> >**REFERENCES**
>
> [1] Zhang, Yichi, et al. "Realsafe-r1: Safety-aligned deepseek-r1 without compromising reasoning capability." arXiv preprint arXiv:2504.10081 (2025).
>
> [2] Jiang, Fengqing, et al. "Safechain: Safety of language models with long chain-of-thought reasoning capabilities." arXiv preprint arXiv:2502.12025 (2025).
>
> [3] Chu, Tianzhe, et al. "Sft memorizes, rl generalizes: A comparative study of foundation model post-training." arXiv preprint arXiv:2501.17161 (2025).
>
> [4] Kirk, Robert, et al. "Understanding the effects of rlhf on llm generalisation and diversity." arXiv preprint arXiv:2310.06452(2023).
>
> [44] Wang, Haoyu, et al. "Safety Reasoning with Guidelines." arXiv preprint arXiv:2502.04040 (2025).

---

> > ### Author Response · Authors · 2025-08-04
> >
> > Please let us know if you had the chance to read through the rebuttals and whether your concerns are addressed. If not, please feel free to ask further questions since the discussion period concludes shortly. Otherwise, we would sincerely appreciate your consideration in raising the score. Thank you!

---

> > ### Comment · Reviewer_EJhw · 2025-08-06
> > **Thanks for your response**
> >
> > The authors have addressed my concerns.
> >
> > I keep my score.

---

> > > ### Author Response · Authors · 2025-08-06
> > >
> > > Thank you for your response! We would greatly appreciate it if you could clarify any additional suggestions that could strengthen our work and potentially lead to a higher score.

---

### Decision · Program_Chairs · 2025-09-17

**Decision:**

Accept (poster)

**Comment:**

This paper proposes an RL-based framework (Training Adaptive Reasoners for Safety or TARS) that uses chain-of-thought to balance safety with task completion in LLMs. In so doing, the authors take a principled approach to using reasoning to defend against harmful prompts where typical defenses use adversarial training or otherwise statistical approaches. A number of practical tricks are employed to prevent the LLM from taking “short cuts” to avoid harmful behavior. Evaluation demonstrates improvements against a range of attacks, promising behaviors on ambiguous queries while preserving utility and indeed improvements to unseen generalization.

Reviewers highlighted strengths of the work, including: quality of writing, thorough ablation studies that help attribute improvements to the many components of the approach. I’ll return below to positives as I saw them.

Some weaknesses were observed, to highlight the key initial ones against clarifications/improvements: how carefully curated the baseline SFT data was and requiring more baselines (additional experiments with three different SFT based methods), limited initial 1.5B model size (expanded to 7B), why adding harmless prompts and separating reward models was necessary (interesting follow-results on response length on ambiguous prompts during training without harmless prompts), that reasoning for safety with traces is not new (however aspects of the approach are), that other works consider general not reasoning models (however reasoning models are important and safety is clearly well motivated).

With all four reviewers sitting on the positive side at the end of the process, but only just (all rating 4s), the decision came down to significance of the work. While I believe this is a borderline case in terms of novelty, on balance I believe the conceptual ideas of the 3-step recipe has potential good impact for the area – it is clear that properly managed CoT has been a relatively untapped advantage of LLMs, and the approaches to manage this reasoning-based defense are elegant, with the particular use of RL and its proper calibration. That said, as LQBK highlighted, conceptually this is not wholly novel but well executed. The results during discussion of the 7B model outperforming SOTA 8B is very promising. Generalization beyond safety applications is compelling. Generally, TARS outperformed baselines on the new experiments requested by reviewers, and while some reviewers didn’t raise their scores at all, or only a little, not raising scores further still was not adequately justified through final justifications or AC-reviewer discussion, or corresponded to reviewers asking relatively tangential questions (total training cost, data complexity for aligning larger LLMs, comparing TARS and deliberative alignment, etc. All interesting but not directly relevant to the assessment in my mind.)